# Plasma Phospholipid Fatty Acids and Risk of Venous Thromboembolism: Mendelian Randomization Investigation

**DOI:** 10.3390/nu14163354

**Published:** 2022-08-16

**Authors:** Shuai Yuan, Xue Li, Pierre-Emmanuel Morange, Maria Bruzelius, Susanna C. Larsson

**Affiliations:** 1Unit of Cardiovascular and Nutritional Epidemiology, Institute of Environmental Medicine, Karolinska Institutet, 17165 Stockholm, Sweden; 2Department of Big Data in Health Science School of Public Health, Center of Clinical Big Data and Analytics of The Second Affiliated Hospital, Zhejiang University School of Medicine, Hangzhou 310058, China; 3Laboratory of Haematology, La Timone Hospital, 13385 Marseille, France; 4Center for CardioVascular and Nutrition Research (C2VN), Institut National de la Recherche Agronomique (INRA), INSERM, Universite Aix-Marseille, 13385 Marseille, France; 5Centre de Ressources Biologiques Assistance Publique–Hôpitaux de Marseille, HemoVasc, 13385 Marseille, France; 6Department of Medicine Solna, Karolinska Institutet, 17177 Stockholm, Sweden; 7Coagulation Unit, Department of Hematology, Karolinska University Hospital, 17165 Stockholm, Sweden; 8Unit of Medical Epidemiology, Department of Surgical Sciences, Uppsala University, 75185 Uppsala, Sweden

**Keywords:** deep vein thrombosis, fatty acid, Mendelian randomization, pulmonary embolism, venous thrombosis

## Abstract

Circulating fatty acids may affect thrombosis but epidemiological data on the associations between fatty acids and risk of venous thromboembolism (VTE) are limited and conflicting. We conducted a Mendelian randomization study to examine the causal associations of 10 circulating fatty acids with VTE risk. Genetic variants strongly associated with ten fatty acids and without linkage disequilibrium were selected as instrumental variables from the Cohorts for Heart and Aging Research in Genomic Epidemiology Consortium. Genetic associations for VTE and its subtypes were obtained from the International Network Against Venous Thrombosis Consortium (30,234 cases and 172,122 controls) and the FinnGen study (11,288 VTE cases and 254,771 controls). Estimates from the two data sources were combined. Per standard deviation increase in genetically predicted fatty acid levels, the combined odds ratio (OR) of VTE was 0.88 (95% confidence interval [CI] 0.84–0.92) for α-linolenic acid, 0.92 (95% CI 0.90–0.95) for linoleic acid, 0.85 (95% CI 0.78–0.92) for palmitoleic acid, 0.77 (95% CI 0.77–0.84) for oleic acid, 1.16 (95% CI 1.10–1.23) for eicosapentaenoic acid, 1.10 (95% CI 1.06–1.14) for docosapentaenoic acid, 1.06 (95% CI 1.04–1.08) for arachidonic acid, and 1.19 (95% CI 1.11–1.28) for stearic acid. Genetically predicted levels of docosahexaenoic acid or palmitoleic acid were not associated with VTE risk. Four and eight out of ten genetically predicted fatty acid levels were associated with risk of pulmonary embolism and deep vein thrombosis, respectively. This study suggests that strategies targeting at fatty acids may act as prevention approaches for VTE.

## 1. Introduction

Venous thromboembolism (VTE), including deep vein thrombosis (DVT) and pulmonary embolism, (PE) is a common medical condition with a high global incidence rate of 1 to 2 persons per 1000 person-years [1]. Risk factors persuasively shown for VTE include increasing age, major surgery, extended immobility, malignancy, multiple traumas, and heart failure [2]. Some modifiable factors, including obesity and lifestyle factors, may also affect the risk of VTE [3,4,5].

With respect to diet, recent observational studies have suggested that dietary intake of polyunsaturated n-3 fatty acids is associated with the risk of onset and reoccurrence of VTE [6,7]. A randomized controlled trial also observed that the group with daily supplementation of omega-3 fatty acids had a lower risk of PE and symptomatic DVT compared to the control group after surgery among patients with proximal femoral fractures [8]. Nevertheless, different individual polyunsaturated n-3 fatty acids may have varying effects on thrombosis and cardiovascular health [9]. Animal studies found that plant-derived α-linolenic acid (ALA) belonging to n-3 fatty acids was against arterial thrombus formation [10,11], tissue factor expression, and platelet activation [11]. The effect of ALA on platelet phospholipids was also observed in vegetarian men [12]. However, another n-3 fatty acid, docosahexaenoic acid (DHA), was not found to be associated with physiological changes in blood coagulation, platelet aggregation, or thrombotic tendencies among healthy adult males [13]. Other fatty acids, such as arachidonic acid (AA) [14], ratio of oleic to palmitic acid (PA) [15], and stearic acid (SA) [16], were also associated with thrombotic features and biomarkers. Nonetheless, few population-based studies have examined the associations between other individual fatty acids, such as palmitoleic acid (POA,) and risk of VTE. In addition, the majority of previous evidence was based on observational studies where the associations were likely to be biased by confounding and reverse causation. Thus, the causality of the associations of fatty acids with risk of VTE remains unclear and entangled.

Mendelian randomization (MR) analysis based on germline genetic data is an epidemiological approach that can strengthen the causal inference in an association by using genetic variants as instrumental variables for an exposure [17]. The method has two advantages over traditional observational studies: (1) it can minimize confounding, since genetic variants are randomly assorted at conception and therefore unassociated with environmental and self-adopted factors; and (2) it can diminish reverse causality as germline genotype cannot be modified by the onset or progression of the disease. Here, we conducted an MR analysis to explore the associations of ten fatty acids with risk of VTE and its two subtypes.

## 2. Methods

### 2.1. Genetic Instrument Selection

Single nucleotide polymorphisms (SNPs) associated with ten plasma phospholipid fatty acids at the genome-wide significance level (*p* < 5 × 10^−8^) were extracted from corresponding genome-wide association analyses in the Cohorts for Heart and Aging Research in Genomic Epidemiology (CHARGE) Consortium [18,19,20]. In detail, SNPs for four n-3 polyunsaturated fatty acids (ALA, eicosapentaenoic acid [EPA], docosapentaenoic acid [DPA], and DHA) were obtained from a genome-wide analysis including up to 8866 individuals of European ancestry [18]. Likewise, SNPs for two n-6 polyunsaturated fatty acids (linoleic acid [LA] and AA) and SNPs for one n-7 monounsaturated fatty acid (POA), one n-9 monounsaturated fatty acid (oleic acid [OA]), and two saturated fatty acid (PA and SA) were selected from a genome-wide analysis including 8631 [19] and 8961 [20] individuals of European ancestry, respectively. The SNP-fatty acid associations were adjusted for age, sex, and study-specific covariates [18,19,20]. SNPs with linkage disequilibrium (i.e., correlated) were excluded and the SNP with the smallest *p* value was retained. SNPs in *GCKR* gene region that shows pleotropic effects with obesity and other metabolic traits were removed, leaving one to four SNPs as instrument variables for above fatty acids (Appendix A). SNPs in or close to *FADS1/2* and *ELOVL2* genes that encode enzymes with a vital role in fatty acid metabolic pathway were associated with several fatty acids (Figure 1) [21]. We rescaled the unit for studied fatty acids levels to one standard deviation (SD) change [21,22].

A supplemental genetic instrument consisting of 9 SNPs without linkage disequilibrium was used for plasma phospholipid AA levels (Appendix A) [19]. These SNPs showed no pleiotropic associations with potential confounders (such as education, Townsend index, smoking, and physical activity) [23]. Given more SNPs included in this genetic instrument, we conducted a sensitivity analysis excluding rs174547 in *FADS1* gene region that explains a considerable variance in AA levels to examine whether the association was driven by this SNP. The genetic associations based on these SNPs were scaled as one SD increment in genetically predicted levels of AA.

### 2.2. Data Sources for Outcomes

Summary level data on the associations of fatty acid-associated SNPs with VTE were obtained from the International Network Against Venous Thrombosis (INVENT) Consortium [24] and the FinnGen study [25]. The INVENT Consortium included 30,234 VTE cases and 172,122 controls from 18 studies with varying diagnostic approaches, which were detailly described in the genome-wide association study [25]. The associations in the INVENT consortium were adjusted for age, sex, array, study-specific covariates, participant relatedness, and top 10 genetic principal components [25]. For the FinnGen study, we used the R6 release data on VTE, pulmonary embolism (PE), and deep vein thrombosis (DVT) from the corresponding genome-wide association analyses of 11,288 VTE cases, 5130 PE cases, 5632 DVT cases, and up to 254,771 controls. Association tests in FinnGen had been adjusted for age, sex, first 10 genetic principal components, and genotyping batch. We used data from two sources which are displayed in Appendix A.

### 2.3. Statistical Analysis

SNPs were harmonized to omit ambiguous SNPs with non-concordant alleles and palindromic SNPs with ambiguous minor allele frequency. The *F* statistic was used to measure the strength of instrumental variables [26]. The inverse variance weighted method with fixed effects was used to estimate MR associations between genetically predicted levels of fatty acids and risk of VTE given a few instrumental variables were used for studied fatty acids. Estimates from the INVENT consortium and the FinnGen study were combined using the fixed effects meta-analysis method due to limited heterogeneity. Sensitivity analyses including the inverse variance weighted method with random multiplicative effects, the weighted median method [27], and MR-Egger regression [28] were conducted to examine the consistency of results for fatty acids proxied by >4 SNPs (POA, and AA proxied by the supplemental instrument). We applied a Benjamini-Hochberg correction to account for multiple testing and the associations with FDR < 0.05 were considered as statistically significant. The association with the *p* < 0.05 and FDR ≥ 0.05 was deemed suggestive. Power was estimated using an online tool (Appendix A) [29]. All of the tests were two-sided and performed using the mrrobust package [30] in Stata/SE 15.0.

## 3. Results

All SNPs were available in the two outcome data sources. The *F* statistics of SNPs for ten fatty acids were over 10, indicating good strengths of employed genetic instruments (Appendix A).

Genetically predicted higher levels of ALA, LA, POA, and OA, and lower levels of EPA, DPA, AA, and SA were associated with a reduced risk of VTE (Figure 2). The patterns of the associations were consistent between the INVENT consortium and the FinnGen study (Figure 2). Per SD increase in genetically predicted levels of fatty acid, the combined odds ratio (OR) of VTE was 0.88 (95% confidence interval [CI] 0.84–0.92) for ALA, 0.92 (95% CI 0.90–0.95) for LA, 0.85 (95% CI 0.78–0.92) for POA, 0.77 (95% CI 0.77–0.84) for OA, 1.16 (95% CI 1.10–1.23) for EPA, 1.10 (95% CI 1.06–1.14) for DPA, 1.06 (95% CI 1.04–1.08) for AA, and 1.19 (95% CI 1.11–1.28) for SA. All of these associations remained statically significant after multiple testing corrections (Appendix A). Genetically predicted higher levels of DHA were suggestively associated with a decreased risk of VTE in FinnGen (OR, 0.76; 95% CI 0.60–0.96) but not in the INVENT consortium (Figure 2). Genetically predicted levels of DHA or PA were not associated with VTE risk in the combined analysis (Figure 2).

Genetically predicted higher levels of ALA, LA, OA, and lower levels of EPA, DPA, and AA showed associations with PE risk (Figure 3). However, the association for LA or OA did not remain after multiple testing corrections (Appendix A). The OR of PE per one SD increase in genetically predicted fatty acid was 0.88 (95% CI 0.79–0.97) for ALA, 1.24 (95% CI 1.07–1.43) for EPA, 1.14 (95% CI 1.05–1.24) for DPA, and 1.07 (95% CI 1.02–1.12) for AA. There were inverse associations of DVT risk with genetically predicted higher levels of ALA, LA, POA, and OA as well as with genetically predicted lower levels of EPA, DPA, AA, and SA (Figure 3). All of these associations persisted after Benjamini-Hochberg correction (Appendix A). Per one SD increment in genetically predicted levels of fatty acid, the OR of DVT was 0.81 (95% CI 0.74–0.90) for ALA, 0.89 (95% CI 0.83–0.96) for LA, 0.65 (95% CI 0.52–0.81) for POA, 0.65 (95% CI 0.53–0.80) for OA, 1.21 (95% CI 1.05–1.39) for EPA, 1.18 (95% CI 1.09–1.27) for DPA, 1.11 (95% CI 1.06–1.16) for AA, and 1.34 (95% CI 1.12–1.59) for SA.

For AA, the associations with VTE, PE, and DVT were consistent in the sensitivity analysis based on the supplemental genetic instrument including and excluding rs174547 in the *FADS1/2* gene region (Figure 4). Notably, the magnitude of the associations became stronger, albeit with wider CI after excluding rs174547 (combined OR for VTE, 1.14; 95% CI 1.08–1.19; OR for PE, 1.14; 95% CI 1.00–1.30; OR for DVT, 1.22; 95% CI 1.08–1.38). The associations for POA and AA proxied by the supplemental genetic instrument were overall consistent in the sensitivity analyses. We observed evidence of potential horizontal pleiotropy in the associations of genetically predicted levels of POA with risk of VTE and DVT in FinnGen (Appendix A).

## 4. Discussion

We conducted an MR study to examine the associations of genetically predicted plasma phospholipid levels of ten fatty acids with risk of VTE, PE, and DVT using data from the INVENT consortium and the FinnGen study. Eight fatty acids were associated with VTE risk, in particular DVT risk, and the pattern of the associations was consistent between two data sources (Figure 5). Overall, genetically predicted higher levels of ALA, LA, POA, and OA, and lower levels of EPA, DPA, AA, and SA were associated with a decreased risk of VTE.

Polyunsaturated n-3 fatty acids in relation to thrombotic events and biomarkers have been studied and discussed in previous studies, even though the findings were conflicting. In 41 patients with hyperlipemia, n-3 fatty acids together with simvastatin reduced the free tissue factor pathway inhibitor fraction and inhibited the activation of factor VII occurring [31] even though statin treatment can also reduce VTE risk [32]. Another trial found that n-3 fatty acids plus standard therapy decreased thrombin formation and favorably altered fibrin clot properties among patients undergoing percutaneous coronary intervention [33]. A recent trial also observed lower risk PE as well as symptomatic DVT in the group with daily supplementation of n-3 fatty acids compared to the control group among patients with proximal femoral fractures [8]. However, the anti-thrombotic impact of n-3 fatty acids was not found in other studies [34,35]. A study including patients with end-stage renal disease found that n-3 fatty acids did not influence the risk of bleeding [34]. Another study in elderly subjects after suffering a myocardial infarction observed that one-year supplementation with n-3 fatty acids did not modulate prothrombotic micro-vesicle release from blood and vascular cells [35]. The discrepancy across above studies might be caused by features of different studied populations and varying effects of individual n-3 fatty acids on thrombotic process. ALA was consistently inversely associated with thrombus formation possibly via lowering tissue factor expression and platelet activation [11]. However, these data were mainly obtained from animal studies. The current MR based on human genetic data confirmed the beneficial effects of high plasma levels of ALA on the VTE development. EPA and DHA were usually studied together and found to have similar effects on anti-thrombosis [36,37]. A diet with a high intake of EPA and a low intake of AA was found to protect against thrombosis [37]. In our study, we found an increased risk of VTE for high plasma levels of EPA and a suggestive decreased risk of VTE for high plasma levels of DHA in FinnGen. The disagreement on EPA might be caused by complex correlations across n-3 fatty acids that might bias the findings in observational studies. For example, high intake of EPA may increase the plasma levels of its upstream and downstream fatty acids, such as ALA and DHA, since high levels of EPA may inhibit the metabolism of ALA (the precursor of EPA) and facilitate the generation of DHA (the metabolite of EPA) that are possibly inversely associated with thrombosis. For DHA proxied by one SNP that explains a small phenotypic variance, the null finding in the INVENT consortium might lack power. There were few studies focusing on the association between DPA and thrombosis. Even though our finding for DPA is consistent with our previous MR finding based on the UK Biobank study [38], more research is needed to verify this novel association.

LA was found to be associated with an improved profile of lipids and blood pressure that show inverse associations with risk of atherosclerotic events [39]. However, limited evidence was collected on the association between LA and thrombosis. Thus, our novel finding on LA in relation to VTE needs to be confirmed, although this association is consistent with our previous finding [38]. On the contrary, AA as another n-6 fatty acid and its metabolites has been studied in relation to thrombosis since the last century [37]. Thromboxane B2 as a major metabolite of AA produced by monocytes contributed to pathogenic thrombosis [40]. Aspirin can bind to cyclooxygenase 1 and 2, which prevents the formation of prostaglandin H2 and subsequently thromboxane and prostaglandin [41], and thus reduce VTE risk [42] (Figure 1). Another study found that residual AA induced platelet activation via several pathways in 700 consecutive aspirin-treated patients undergoing cardiac catheterization [14]. A recent review discussed possible mechanisms of atherothrombosis caused by AA metabolism [43]. Even though certain metabolites of AA, such as prostanoids, have been linked to thrombosis, the association between AA and thrombotic disease has been less extensively explored in population-based studies [43]. Our study based on two sets of genetic instruments as well as two independent populations suggested that high levels of plasma AA levels were associated with increased risk of VTE and its subtypes, which strengthened the vital role of AA-related pathways in the pathophysiology of thrombosis.

A few studies have been conducted to examine the associations of POA, OA, and PA with thrombosis. A study of 14 healthy men found that ratio of OA to PA was inversely associated with postprandial concentrations of tissue factor, fibrinogen, and plasminogen activator inhibitor 1 [15] that are associated with VTE risk [44]. Even though our MR analysis did not observe an association between PA and VTE risk, the inverse association between OA and VTE risk supported the above finding. Several studies examined the association of SA-rich diet with thrombotic features and biomarkers. Contradictory to the observed positive association between SA and VTE risk, SA-rich diet was not associated with an increased risk of thrombosis or higher levels of thrombotic biomarkers [16,45,46]. SA was found to have similar effects as OA and LA on markers of thrombotic tendency in healthy human subjects [47]. However, given that SA can be derived from a wide range of dietary sources including meat, poultry, fish, milk, dairy products, and cocoa butter, the anti-thrombosis nutrients from the SA-rich diet might introduce bias in the observational findings. Thus, the effects of SA on venous thrombosis need further investigation.

A strength of this study is the MR design, which minimizes confounding as well as reverse causality. Another strength is the large sample sizes of two independent data sources that allowed for an exploration of modest associations. Several limitations deserve considerations when interpretating our findings. Firstly, there were approximately 4.4% sample overlap for the analysis in the INVENT consortium, which might make the analytic model overfitting when the genetic instruments are weak (the weak instrument bias) [26]. However, the *F* statistic was used to measure the strength of used instrument and all *F* statistics over 10 indicated good strength of employed genetic instruments and no weak instrument bias. Secondly, even though we used two data sources with larger sample sizes and combined associations, we might overlook some weak associations due to an inadequate power that was caused by a small phenotypic variance explained by a few genetic variants, such as the association for DHA. Thirdly, SNPs in *FADS1/2* gene region are strongly associated with several fatty acids and explain the majority of variance in corresponding fatty acids [21,38]. Thus, we could not expel the mutual influences across fatty acids partly proxied by this shared genetic locus although the mutual influences make biological sense since *FADS1/2* gene encodes key enzymes, such as delta-6 and delta-5 desaturase enzymes, in the metabolism of fatty acids [21]. To examine this hypothesis, we used a supplement set of more genetic variants for AA and excluded SNPs in *FADS1/2* gene in a sensitivity analysis. The consistent association from this sensitivity analysis indicated the robustness of associations in the main analysis. Fourthly, pleiotropic effects could not be assessed using MR sensitivity analyses. However, used SNPs appeared to be associated with a few pleiotropic traits that might bias the observed associations [21,23]. Fifthly, our study focused on plasma levels of fatty acids that do not necessarily equate to dietary intake. More studies are needed to examine the association between dietary intake and circulating levels of fatty acids. Sixthly, we could not analyze the influence of characteristics of the study populations on our findings due to the lack of individual-level data. In addition, some hypolipidemic drugs, such as statins, can alter the levels of serum fatty acids [48]. However, the lack of medication data confines the analysis on how hypolipidemic drugs generate effects on our results.

## 5. Conclusions

In conclusion, this MR study found that genetically predicted higher levels of ALA, LA, POA, and OA, and lower levels of EPA, DPA, AA, and SA were associated with the decreased risk of VTE, in particular DVT. Our findings suggests that strategies targeting fatty acids may act as prevention approaches for VTE. For example, increasing intake of vegetable oils from soybean, olive, canola, and sunflower, and decreasing intake of animal products may lower the risk of developing VTE. In addition, drugs that can alter levels of identified fatty acids may also be used to prevent VTE.

## Figures and Tables

**Figure 1 nutrients-14-03354-f001:**
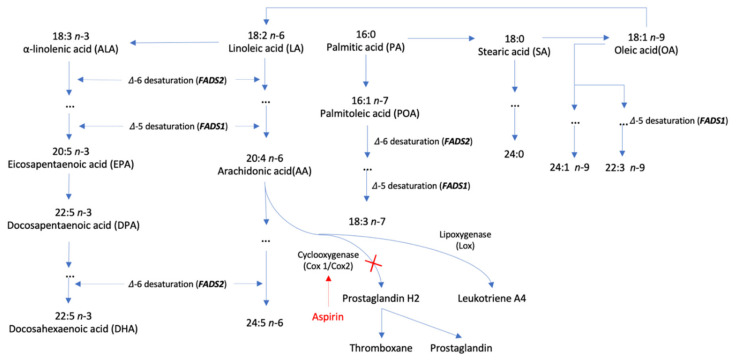
Metabolism of fatty acids and its relation to aspirin therapy.

**Figure 2 nutrients-14-03354-f002:**
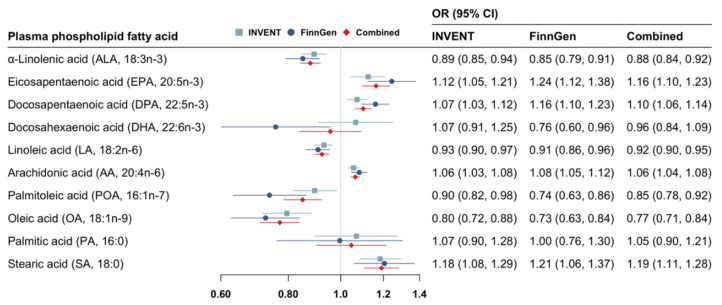
Associations of genetically predicted circulating levels of fatty acids with risk of venous thromboembolism. CI indicates confidence interval; INVENT, the International Network Against Venous Thrombosis; OR, odds ratio. The associations were scaled to one standard deviation increment in genetically predicted circulating levels of fatty acids.

**Figure 3 nutrients-14-03354-f003:**
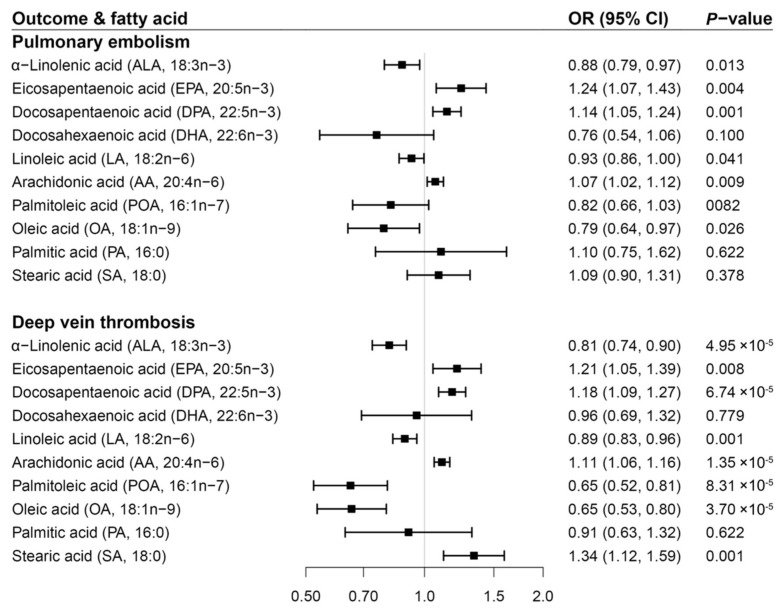
Associations of genetically predicted circulating levels of fatty acids with risk of pulmonary embolism and deep vein thrombosis in the FinnGen study. CI indicates confidence interval; OR, odds ratio. The associations were scaled to one standard deviation increment in genetically predicted circulating levels of fatty acids.

**Figure 4 nutrients-14-03354-f004:**
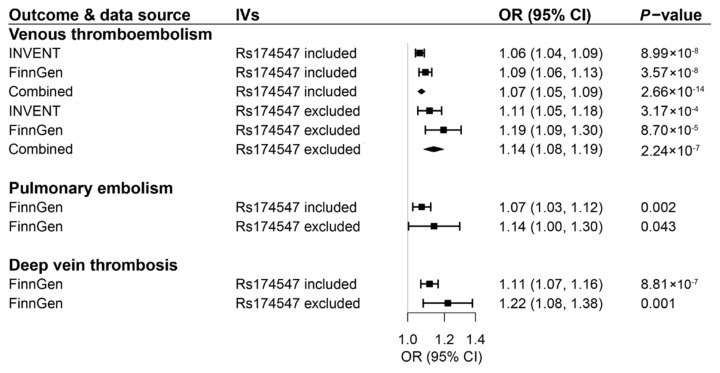
Associations of genetically predicted circulating levels of arachidonic acid based on a supplemental genetic instrument consisting of 9 or 8 SNPs with risk of venous thromboembolism and its two subtypes. CI indicates confidence interval; IVs, instrumental variables; OR, odds ratio. The associations were scaled to one standard deviation increment in genetically predicted circulating levels of fatty acids.

**Figure 5 nutrients-14-03354-f005:**
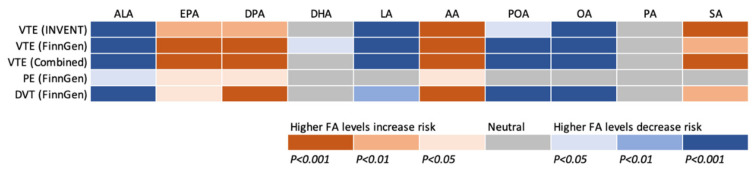
Summary of associations of genetically predicted levels of ten fatty acids with risk of venous thromboembolism and subtypes after correction for multiple testing. AA indicates arachidonic acid; ALA, α-linolenic acid; DHA, docosahexaenoic acid; DPA, docosapentaenoic acid; DVT, deep vein thrombosis; EPA, eicosapentaenoic acid; FA, fatty acid; LA, linoleic acid; INVENT, the International Network Against Venous Thrombosis; OA, oleic acid; PA, palmitic acid; PE, pulmonary embolism; POA, palmitoleic acid; SA, stearic acid; VTE, venous thromboembolism.

## Data Availability

Data used were attached in the Appendix A.

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
