# Peer review of "Plasma Phospholipid Fatty Acids and Risk of Venous Thromboembolism: Mendelian Randomization Investigation"

_nutrients, 2022, doi:10.3390/nu14163354_

Round 1
Reviewer 1 Report
Dear Editor,
I really appreciate the paper from Yuan et al. which dealt with the identification of possible correlation between thrombosis and circulating fatty acids. The idea to identify specific genetic variants related to fatty acids synthesis and composition is intriguing. Some points should be addressed:
1. VTE are due to several conditions more ofter different from "lipidic profile" of individuals. The authors did not comment about the possible key factors for the occurrence of VTE. The clinical variability might impact on results and reduce the strenght of the conclusions.
2. similar considerations are according to the thrombophilic profile of individiduals. The genetic variations in thrombophilic risk might me more related to the occurrence of VTE than circulating fatty acids. This is a key point for this paper.
3. The lack of description of the characteristics of the study population is still a matter of debated for this type of study. This should be deeply discussed.
Author Response
Reviewer 1:
I really appreciate the paper from Yuan et al. which dealt with the identification of possible correlation between thrombosis and circulating fatty acids. The idea to identify specific genetic variants related to fatty acid synthesis and composition is intriguing. Some points should be addressed:
Response to the comment: We sincerely thank the Reviewer for reviewing our paper and providing constructive comments. We have now carefully considered the comments and revised the manuscript accordingly.
- VTE are due to several conditions more often different from "lipidic profile" of individuals. The authors did not comment about the possible key factors for the occurrence of VTE. The clinical variability might impact on results and reduce the strength of the conclusions.
Response to comment 1: We thank the Reviewer for raising this insightful point. We agree that the lipidic profile may play a role in the occurrence of VTE. Intake of different fatty acids has been found to influence the levels of lipids (1-3), which indicates that lipidic profile may mediate the associations between fatty acids and VTE risk. A few pleiotropic traits have been found to be associated with used genetic instruments for fatty acids except for rs174547 in FADS1 gene in our previous study (4), which implies that the current results were less likely to be biased by pleiotropy possibly caused by varying clinical features.
Refs:
- Mensink RP, Zock PL, Kester AD, Katan MB. Effects of dietary fatty acids and carbohydrates on the ratio of serum total to HDL cholesterol and on serum lipids and apolipoproteins: a meta-analysis of 60 controlled trials. Am J Clin Nutr. 2003;77(5):1146-55.
- Hartweg J, Farmer AJ, Perera R, Holman RR, Neil HA. Meta-analysis of the effects of n-3 polyunsaturated fatty acids on lipoproteins and other emerging lipid cardiovascular risk markers in patients with type 2 diabetes. Diabetologia. 2007;50(8):1593-602.
- Laidlaw M, Holub BJ. Effects of supplementation with fish oil-derived n-3 fatty acids and gamma-linolenic acid on circulating plasma lipids and fatty acid profiles in women. Am J Clin Nutr. 2003;77(1):37-42.
- Yuan S, Larsson SC. Association of genetic variants related to plasma fatty acids with type 2 diabetes mellitus and glycaemic traits: a Mendelian randomisation study. Diabetologia. 2020;63(1):116-123.
- Similar considerations are according to the thrombophilia profile of individuals. The genetic variations in thrombophilia risk might be more related to the occurrence of VTE than circulating fatty acids. This is a key point for this paper.
Response to comment 2: The associations of genetic instruments with fatty acids were much stronger than that with VTE risk even though the genome-wide association studies on VTE included a larger number of participants. Please refer to the data that are displayed in supplementary table 3 and 4.
- The lack of description of the characteristics of the study population is still a matter of debated for this type of study. This should be deeply discussed.
Response to comment 3: We agree that the lack of data on the characteristics of the study population is a limitation of the study. We have now discussed this as a limitation in the manuscript.
Page 8:
“Sixthly, we could not analyze the influence of characteristics of the study populations on our findings due to the lack of individual-level data.”
Reviewer 2 Report
I would like to sincerely thank the authors for the study. Please, do not consider the following questions badly - I only suppose that the answers to them may be interesting for the further evaluation of the data from the manuscript:
-
It is known that VTE has more risk factors, such as the inflammatory activity, presence of oncological disease, further comorbidities of the patients, trauma, surgery, pregnancy and postpartum period, hormonal substitution treatment, etc. Therefore, can the authors add the clinical data about the participants in the study ?
-
The levels of fatty acids can be modifiable by the treatment with hypolipidemic drugs (statins, fibrates...). Can the authors clarify whether the included participants took such drugs – it could influence the results...
-
From the practical point of view, can the authors make a summary of their recommendations based on the results of this study ?
From my perspective, the article can be published after incorporation of the answers to the questions and after such a minor revision.
Author Response
Reviewer 2:
I would like to sincerely thank the authors for the study. Please, do not consider the following questions badly - I only suppose that the answers to them may be interesting for the further evaluation of the data from the manuscript:
Response to the comment: We appreciate the Reviewer for reviewing our paper and the positive comments. We have now carefully considered these comments and made corresponding revisions to the manuscript.
- It is known that VTE has more risk factors, such as the inflammatory activity, presence of oncological disease, further comorbidities of the patients, trauma, surgery, pregnancy and postpartum period, hormonal substitution treatment, etc. Therefore, can the authors add the clinical data about the participants in the study?
Response to comment 1: We sincerely thank the Reviewer for pointing this out. Unfortunately, we are not able to add data on the above-mentioned factors given that our analysis was based on summary-level data. We have now discussed this as a limitation in the manuscript.
Page 8:
“Sixthly, we could not analyze the influence of characteristics of the study populations on our findings due to the lack of individual-level data.”
- The levels of fatty acids can be modifiable by the treatment with hypolipidemic drugs (statins, fibrates...). Can the authors clarify whether the included participants took such drugs – it could influence the results...’
Response to comment 2: This is an interesting point. Unfortunately, the lack of medication data confines further analysis on how hypolipidemic drugs generate effects on our results. We have now discussed this point in the manuscript.
Page 8:
“In addition, some hypolipidemic drugs, like statins, can alter the levels of serum fatty acids [48]. However, the lack of medication data confines the analysis on how hypolipidemic drugs generate effects on our results.”
- From the practical point of view, can the authors make a summary of their recommendations based on the results of this study?
Response to comment 3: We have now added a summary of recommendations on the intake of vegetable oils and animal products based on our findings as suggested.
Page 8:
“For example, increasing intake of vegetable oils from soybean, olive, canola, and sunflower, and decreasing intake of animal products may lower the risk of developing VTE. In addition, drugs that can alter levels of identified fatty acids may also be used to prevent VTE.”
From my perspective, the article can be published after incorporation of the answers to the questions and after such a minor revision.
Response to the comment: Many thanks for the positive feedback.